# “We Work in an Industry Where We’re Here to Care for Others, and Often Forget to Take Care of Ourselves”: Aged-Care Staff Views on Self-Care

**DOI:** 10.3390/geriatrics10010003

**Published:** 2025-01-02

**Authors:** Anna P. Lane, Jennifer Tieman

**Affiliations:** 1National Ageing Research Institute, 34-54 Poplar Road, Gate 4, Building 8, Royal Melbourne Hospital, Parkville, VIC 3050, Australia; a.lane@nari.edu.au; 2Research Centre for Palliative Care, Death and Dying, Palliative and Supportive Services, College of Nursing and Health Sciences, Flinders University, Adelaide, SA 5042, Australia

**Keywords:** self-care, residential aged-care facility, terminal care

## Abstract

**Objective**: This study aimed to explore self-care understanding and behaviours among aged-care workers in Australia. It was conducted as part of a project to co-produce a self-care resource for the Australian aged-care workforce. **Methods**: Semi-structured interviews with eleven aged-care staff and a focus group with four staff at an aged-care facility were undertaken to understand how staff understand and practice self-care and how death and dying affect workers. Thematic analysis was performed using software to generate a data coding tree. **Results**: Aged-care workers view self-care as taking care of oneself and as being a way to manage and maintain wellbeing so that they can continue to care. As practiced in daily life, self-care is highly individualised, with actions at work and at home having significant impact on a person’s wellbeing. Supportive organisational cultures and collegial teams were found to be particularly relevant in helping staff to deal with death and dying. **Conclusions**: Aged-care workers may benefit from an online self-care resource tailored to their specific needs and based on their first-hand experiences of working in aged-care. Organisations can support aged-care workers by creating space and time for self-care.

## 1. Introduction

The purpose of this study was to explore the self-care experiences of aged-care workers—defined as those providing services to older people in aged-care facilities and the home—to inform the development of an online support resource. The importance of aged-care has increasingly been recognised as the world’s population ages and the need for care and support in older years grows. Workforce pressures have been acknowledged as well as the changing demands on those providing direct hands-on care, which can lead to fatigue and emotional stress. This is noted in both residential aged-care settings and in those providing aged-care services to clients in the home [1,2,3,4].

Self-care broadly refers to the actions that an individual takes in the interest of their own wellbeing, and often involves personal coping strategies [5]. It is often viewed as a strategy to protect healthcare workers against burnout and compassion fatigue due to occupational stress. Burnout is a “psychological syndrome emerging as a prolonged response to chronic interpersonal stressors on the job” [6] and is characterised by emotional exhaustion, distancing from patients and colleagues, as well as negative feelings and cynicism about one’s job, and decreased motivation and productivity. Compassion fatigue is a reduction in the ability to empathise due to physical and emotional exhaustion, and contributes to burnout [7]. 

The challenges of aged-care work are well documented and include excessive workloads, time constraints, limited career growth opportunities, long working hours, heavy physical demands, insufficient training to deal with complex conditions, and work schedule demands [8,9,10]. For home care workers, concerns include working in unsanitary conditions, working with aggressive patients, and working alone. With an estimated 35% of all deaths in Australia occurring in residential aged-care facilities, dealing with death and dying is considered another source of stress among the aged-care workforce [11]. However, this association is currently underexplored. 

The COVID-19 pandemic has significantly impacted the aged-care workforce. The Royal Commission into Aged Care Quality and Safety concluded in a special report: “The COVID-19 pandemic has been the greatest challenge Australia’s aged care sector has faced. Those who have suffered the most have been the residents, their families and aged care staff” [12]. As frontline workers in an essential industry, aged-care workers reported to their place of work during lockdowns and quickly accommodated new workplace and care protocols. They faced a plethora of anxiety-provoking issues including fatigue from long shifts spent wearing personal protective equipment (PPE), fear of being exposed to COVID-19 at work and taking the infection home to families, as well as anxiety associated with high rates of death in aged-care facilities [13,14,15].

While there is no evidence on the prevalence of burnout among aged-care workers in Australia, a systematic review of burnout in healthcare professionals providing palliative care reported prevalence rates ranging from 3% to 66%, with professionals in general settings experiencing more symptoms of burnout than those in specialist palliative care settings [16]. Aside from the toll on workers’ own health and wellbeing, burnout is harmful to the aged-care sector because it causes high staff turnover and workforce shortages [16,17], and lower-quality patient care [18]. While self-care may be viewed as an individual action and thus an individual responsibility, there are implications for organisations and how they create the conditions in which self-care practice may be normalised and enabled. Self-care theory and frameworks recognise wider determinants of individual self-care behaviours [19].

The importance of self-care for the aged-care workforce is observed in policies and standards governing the sector. Under Standard 7 of the Aged Care Quality Standards, aged-care providers are required to employ a workforce that is “skilled and qualified to provide safe, respectful and quality care services” [20]. Additionally, Standard 9 of the National Palliative Care Standards appeals to employers to support workers with self-care: “Staff are trained in self-care strategies and advised on how to access personal support” [11].

Evidence about self-care and the aged-care workforce is limited. The associations between self-care, wellbeing, and burnout have been examined more extensively in other populations. A number of studies have evaluated self-care planning [21,22], mindfulness-based programmes [23,24,25], yoga [26], and expressive writing [27] in cohorts of nurses working in settings other than aged-care, such as oncology, hospices, and hospitals. In a systematic review of burnout intervention studies for nursing home staff, three Australian studies were identified which all looked specifically at improving care for clients with dementia by supporting dementia care workers [28]. Additionally, knowledge about how self-care influences wellbeing and theory for self-care is underdeveloped [19,29].

The capacity of Australia’s aged-care workforce to deliver quality care to people at the end of their lives is a core ambition of the End of Life Directions for Aged Care (ELDAC) project. Commencing in 2017 and supported by the Australian Government Department of Health, ELDAC is conducted by a national consortium of eight partners including Flinders University. The ELDAC team work to support aged-care workers through provision of tailored and accessible information and resources. Given the evidence linking self-care activity with the capacity to care for others, as well as the stressors facing the aged-care sector currently, ELDAC conducted a project to explore and support self-care as viewed by aged-care staff who provide direct care to older Australians in residential and home care settings. Evidence on self-care among this population is lacking, yet it is imperative to the development of tailored resources to support this vital workforce. Given that this study was conducted during the COVID-19 pandemic, we needed to adjust strategies for participation to meet the social distancing restrictions and lockdowns. In particular, face-to-face focus groups were not feasible, and online interviews and focus groups were used in their place. However, even with these restrictions, through the discovery stage of this project to design a useful and accessible self-care resource, we gained insights into the self-care experiences and needs of aged-care workers. 

## 2. Methods

### 2.1. Participants and Data Collection

The findings presented in this paper represent the first phase of a co-design project conducted online during the COVID-19 pandemic, to develop an online self-care resource. This phase of the co-design process—which engaged aged-care staff to explore their views on self-care—involved ideation. Using insights and knowledge gathered in the first phase, the research team identified initial design elements for an online tool and engaged again with the aged-care staff to seek their inputs and refine further. A prototype (i.e., a web page, with sub-pages of content informed by the findings) was then developed, followed by testing and evaluation.

Participants were working aged-care staff with experience providing direct care to older people in residential aged-care facilities and home care settings in Australia. A convenience sample was recruited through ELDAC marketing channels, which included items in the monthly newsletter and items on ELDAC’s social media channels such as Twitter/X, LinkedIn, and Facebook. Interested staff completed an online expression-of-interest form. The method of approach was open and exploratory, seeking to capture diverse perspectives and include staff across a range of aged-care roles. Demographic data were not obtained.

Data were collected by the first author via semi-structured interviews with eleven staff, and one focus group with four staff at a private residential aged-care facility during a lockdown due to the COVID-19 pandemic. With the intention of co-producing an online resource, during this discovery stage of the project, participants were asked questions covering three core topics: (i) What is ‘self-care’ and why does it matter to aged-care staff? (ii) What resources have been developed for aged-care staff, and can staff benefit from them? (iii) What are some ideas for managing stress and maintaining wellbeing over the long term while doing this work? (to view Interview Guide, see Appendix A). 

Participants were informed of the voluntary and confidential nature of their involvement, and that they could withdraw or decline to answer questions at any time. The interviews and focus group were conducted online between June and August 2021 and were each about 45 to 60 min. They were recorded and transcribed verbatim. All participants received an e-voucher.

### 2.2. Analysis

The analysis was tailored to the needs of the project and the objective of creating a self-help tool for those working in aged-care. Data were thematically analysed (AL) through key stages of familiarisation, reviewing of transcripts, and coding for emergent themes [30]. NVivo software version 12 was used to create a data coding tree. Analysis was deductive and inductive and used to generate material for an online self-care resource. To ensure fidelity between the interview data and emergent themes, the authors met regularly to discuss key findings. Quotations are included here to illustrate the themes identified in the data. Occupations are used to protect participant anonymity. Findings were discussed at project advisory group meetings. The project advisory group comprised stakeholders with practical or subject matter expertise in the areas of aged-care, palliative care, self-care, and technology.

### 2.3. Ethics

This study was conducted as part of a project to co-produce a self-care resource for the Australian aged-care workforce. Ethical approval was obtained from the Flinders University Human Research Ethics Committee (project no. 4652). 

## 3. Results

Participants were 14 aged-care staff, most (*n* = 13) of whom were female and working in the residential aged-care setting. Only one participant worked in the home care setting. The sample included a range of occupational roles that make up the direct care workforce (Table 1). Some (*n* = 4) participants had managerial responsibilities. One person participated in both an interview and a focus group. Key themes are identified in Table 2 and discussed below. 

### 3.1. Holistic Self-Care and Shared Responsibility 

Participants typically used the expression “looking after yourself” to explain the meaning of self-care and connected their health and wellbeing with their capacity to care for others.
“Self-care means looking after yourself in all respects of your health, whether your mental health, your physical health, your spiritual health. I think it goes, the old cliche saying, that you need to fill your cup before you can help others, so making sure that you look after yourself first.”(HSW)

In addition, participants described self-care as taking care of each other, suggesting a collective identity and sense of responsibility for each other’s wellbeing that has a flow-on effect for aged-care residents/clients.
“You’ve got to support each other. You can’t turn around and say, ‘Oh well, that carer can look after herself’, or whatever. You’ve all got to look after each other. If you see that one’s doing more than what they should be doing, step in and you help them.”(PCW)

### 3.2. Self-Care, Individual Responsibility, and Organisational Support

Participants felt self-care was important but something they often overlooked or let slip because the core business of aged-care is to care for others. They frequently described how the nature of the job made it difficult for them to care for themselves.
“We work in an industry where we’re here to care for others, and often forget to take care of ourselves. In the different settings, more so the environment we live in right now, it’s stressful. The job is stressful.”(RN)

Participants viewed self-care as a responsibility of the individual. However, their responses strongly supported a view of organisations as having a role in encouraging and supporting employee self-care, since many of the factors that affect them are beyond their control.
“I think that yes, it still comes back to the individual and you could provide all the tools possible, but the individual’s got to make the choice to use them. But that being said, being a workforce in a changing environment, where we have to give a lot of ourselves, not just physically, emotionally as well, that the organisation should have some part in assisting them to take self-care.”(RN)

### 3.3. Preventing Burnout in the Face of Acute and Ongoing Challenges 

Self-care was typically described as an approach to prevent exhaustion and burnout, and participants identified several work challenges that would lead them to cut back on their hours or leave the workforce completely. Participants described how the ongoing threat of the COVID-19 pandemic, coupled with the aged-care reforms, has created an ever-changing and demanding work environment. One participant described the impact of lockdowns on staff and consumers, as well as their families.
“It’s very hard in aged-care. A lot of people don’t understand, even though we’ve been doing them for 12 months. We’re currently in a lockdown now and the staff and the residents, we went through a terrific gastro outbreak where the facility was locked down. We had one day, we got the clear, we came out of that, and after one day we were put into lockdown for COVID-19 prevention. The last month has been very stressful and tiring for the residents and for the staff, and the families.”(RN)

Another participant described the impacts on staff of change due to reforms and pandemic conditions.
“On top of COVID, and as we keep saying to people, give us a break because we have just lived through the worst two years of probably all our days. And then on top of that, post the Royal Commission is just change after change after change. That’s fatiguing.”(RN/CE)

They described high workloads, the demands of working in dementia care, and a growing weariness with the negative press around the sector as having a significant impact on workers. For managers, recruitment and retention of staff were sources of stress. 

### 3.4. Personalised Approaches to Self-Care and the Role of Workplace Relationships

Participants described many ways in which they were taking care of themselves and others, supporting the view that self-care is different for everyone and there is no one way to look after yourself. When described how they take care, participants identified things that they do at home and work. At work, the quality of relationships and interactions with colleagues and residents were frequently described as having an impact on how they feel. In view of the broad, individualised nature of self-care and participants’ emphasis on the relationship between caring for oneself and caring for one’s colleagues, it was clear that the quality of workplace relationships was a significant aspect of self-care from the perspective of participants. Table 3 shows some behaviours that impact aged-care staff wellbeing. 

### 3.5. Emotional Impact of Death and Dying: Coping Through Care, Connection, and Reflection

Participants reported being affected by the death and dying of the people that they personally cared for. As one personal care worker said:
“Because I try and build a relationship with the people I look after, when they’re palliative and they’re at the end of their life, shedding some tears for them is a way for me of saying I loved her or him and I’m going to miss them.”(CW)

More intense emotional reactions were associated with first death experiences, unexpected or traumatic deaths, hospital deaths, and the presence of a personal connection. As one participant mentioned:
“If people die in hospital, that’s another grief. Because they don’t get to say goodbye. Often the hospital won’t tell you…. The family will tell you first that the person’s died, not the hospital.”(RN/CE)

Death and dying was also seen as a stressor, and one that has been accentuated by the pandemic and aged-care reforms.
“On top of COVID, on top of reform, there is the day-to-day business which in aged-care is all about essentially living until death and dying is your companion. And with the reforms that have come through, death and dying comes much sooner than it used to do in the past. People could stay 5, 8, 10 years in aged-care. Now they come in for usually less than two years. Your churn through palliation is much greater. Your ability to burnout is pretty high….”(RN/CE)

Participants described experiences that take place prior to death and when the resident/client is at the end of life, immediately before and immediately after death, and following death. For most participants, making residents comfortable was a way for them to deal with death and dying.
“And I take a lot of pride in making them comfortable and knowing that I’ve had a big part of that. So that if I am there when they do go, I’m usually holding their hand, or talking to them, “It’s okay. It’s okay.” … I like to take a trolley into the person’s room, we’ve got the mouth swabs on it, so we can keep them moist. I have creams, and I’ve got a music player. And I usually put on their favourite music, or I’ll do the things that they wanted for their end of care if it’s requested.”(PCW)

Participants also recognised the importance of reflection and of expressing emotion to deal with the death of a resident. Indeed, in so far as exposure to death and dying had an impact on the mental wellbeing of aged-care workers, connecting with residents and reflecting on their relationships and experiences with them comprised a key element of self-care. 

## 4. Discussion

This study explored the concept of self-care as it is understood and practiced by aged-care staff in Australia. People working in aged-care understand self-care as taking care of oneself. This is largely consistent with how other healthcare occupational cohorts describe and make sense of self-care, including in the context of palliative care [31]. Self-care may have had particular significance given that the study was conducted during the COVID-19 pandemic, which led to additional workforce stressors and a deeper contact with potential illness and death. Additionally, this personal understanding held by aged-care staff is reflected in the resources that have proliferated because of the COVID-19 pandemic and are available for aged-care workers. These include the web-based resource produced by Phoenix Australia [32], which was one of four organisations funded to deliver an A$12.4 million Grief and Trauma Response Package to the aged-care sector, and the self-care planning tool developed by Palliative Care Australia [33].

This study showed that self-care can mean many things and is highly individualised. This can be seen in the diverse array of practices and behaviours that people have engaged in to take care of themselves and others, and perhaps also reflects workforce diversity. Therefore, there is no one-size-fits-all approach to supporting self-care. For self-care to be effective and to have impact, people will benefit from having access to a range of resources that they can personalise. Ideally, a useful resource will help to connect workers and provide a mechanism for the sharing of ideas from those with first-hand experience. The results of this study point to the importance of providing contextualised supports for aged-care staff, and for designing a useful and accessible resource with aged-care staff, as the need knowers and end users. The messages that resonate are those that capture the stories and experiences of aged-care staff, as told in their own voices. To personalise self-care, aged-care workers are likely to need access to a suite of resources.

The link between self-care and care quality as well as potential workforce sustainability was evidenced, supporting investments by organisations in enabling and fostering opportunities for staff to take up and embed practices of self-care. Promotion of staff wellbeing requires a whole-of-system approach. While individuals have a responsibility for their own wellbeing, organisational responses (such as supervisor support and the development of adaptive coping skills) can facilitate recognition of workplace pressures and enhance self-care [34,35]. This is particularly important given the increasing numbers of older Australians who will be cared for through to the end of life in residential aged-care. A workplace culture that recognises this reality, provides mechanisms for staff to take care of themselves as care providers for older people at the end of life, and destigmatises the expression of emotion is needed.

The themes presented here will be used to orient the resource to the specific needs of its intended users. These themes inform the content of the resource and how best to deliver that content to the end user. For example, the online resource may provide an interactive and dynamic space for aged-care staff to share their own self-care tips and provide a mechanism for staff to support each other with self-care. In recognition of the view that self-care is a shared responsibility, content will target organisations and provide examples of how they can support their managers and teams to practice self-care and look after each other. Additionally, there will be content that raises staff awareness of situations that may impact the wellbeing of staff, such as death and dying, the potential benefits of self-care for maintaining wellbeing and resilience, and encouragement of self-care behaviour via the provision of good practice examples.

### Limitations and Future Research

Given that this study was conducted during the COVID-19 pandemic, insights on how aged-care staff conduct self-care and many of the results may not be applicable after the pandemic, which is a limitation of this study. We note that the absence of demographic data could limit the generalisability of the findings and that this adds to the complexity of considering the impacts on the understanding of self-care and the subsequent development of self-care resources. While we tried to capture a diversity of perspectives, the study would have benefited from greater participation of people from culturally and linguistically diverse (CALD) backgrounds, and from workers in home care. The utility and accessibility of existing resources for the aged-care workforce will need to be addressed in future studies. Care workers are lightly trained, and increasingly the workforce is made up of foreign-born workers with differing levels of English language proficiency. In a multicultural workforce, there are likely to be considerable differences between people in their attitudes and behaviours towards death and dying as well as self-care. Further, care workers in communities, as opposed to residential care settings, face additional occupational stressors and barriers to self-care practice, which a self-care resource will need to take into consideration. In community settings, workers are tasked with meeting the daily, personal care needs of older people while they are living at home. Workers must navigate social complexities involving older peoples’ family and friend networks, and they have limited face-to-face contact with peers, who can be an important source of emotional as well as instrumental support. In residential care settings, workers operate in proximity to their peers, which affords greater opportunities for incidental interaction and mutual support, including with self-care activity. Theory on self-care and the mechanisms via which wellbeing is influenced remains to be resolved. 

## 5. Conclusions

This study found that the aged-care workers who participated viewed self-care as taking care of themselves, as well as a way to manage and maintain wellbeing so that they could continue to provide care. Self-care was viewed as highly individualised, and supportive organisational cultures and collegial teams were found to help staff deal with death and dying. The findings indicate that organisations can support aged-care workers by creating space and time for self-care, and that aged-care workers may benefit from an online self-care resource tailored to their specific needs and based on the first-hand experiences of those working in aged-care. 

## Figures and Tables

**Table 1 geriatrics-10-00003-t001:** Details of participants.

	Occupation	Setting	State	Interviews	Focus Group
1.	Personal care worker (PCW)	RAC	SA		
2.	Home support worker (HSW)	Home	SA		
3.	Personal care worker	RAC	SA		
4.	Consultant in residential aged-care and former personal care worker	RAC	SA		
5.	Registered Nurse (RN), Residential Manager	RAC	QLD		
6.	Chaplain	RAC	NSW		
7.	Chaplain	RAC	NSW		
8.	Pastoral carer	RAC	VIC		
9.	Clinical Nurse Specialist (CNS) (Community palliative care, RACF outreach)	RAC	NSW		
10.	Lifestyle assistant	RAC	SA		
11.	Registered Nurse, Chief Executive (CE)	RAC	ACT		
12.	Clinical manager	RAC	ACT		
13.	Clinical manager	RAC	ACT		
14.	Consumer concierge officer	RAC	ACT		

Note. RAC: residential aged-care. Home: home care. ACT: Australian Capital Territory. SA: South Australia. QLD: Queensland. NSW: New South Wales. VIC: Victoria. The grey shading indicates participation in an interview, a focus group, or both.

**Table 2 geriatrics-10-00003-t002:** Key themes.

	Theme
1.	Holistic Self-Care and Shared Responsibility
2.	Self-Care, Individual Responsibility, and Organisational Support
3.	Preventing Burnout in the Face of Acute and Ongoing Challenges
4.	Personalised Approaches to Self-Care and the Role of Workplace Relationships
5.	Emotional Impact of Death and Dying: Coping Through Care, Connection, and Reflection

**Table 3 geriatrics-10-00003-t003:** Self-care behaviours of aged-care staff.

Actions	Illustrative Quote
**Actions performed at work**	
Going for a walk	I would go for a walk in my lunch break, morning tea break it would be a walk around the block. We used to do walking meetings. (Consultant/Former PCW)
Getting to know colleagues	I like to talk to people. I find out about them. You know, what do they like to do? Do they have family? Like how’s their weekend. Or anything that I can help them with, you know any work-related stuff that I can help them with. You know? … Because you can detect, if one day they become very stressed. They get a bit worked up. You know, this is not you. This is unlike you. (CNS)
Debriefing	Debriefing, that’s so important. And you know when someone debriefs to you, you’re not judging them. You’re giving them, “What have you done well. Maybe next time, you know this is what we can improve. But that’s all right. You know I’m not blaming, but let’s learn from this”. (CNS)
Getting to know residents	Probably something that’s made me feel good in the work setting is having time with the residents. It’s not task focused, it’s not going to see them because of a complaint. It’s genuinely sitting with them and learning their story or speaking with them as a fellow adult, and not being a nurse and having to go to them because you have to, but learning from them … . (RN)
Releasing emotion	I help a gentleman once a fortnight. I go around, and we take turns, there’s different carers that go, but once a fortnight, it’s my job to go and help him with his personal care. And he has a brain tumour, and he’s fairly severely impacted by that. So it’s tough. It’s physically tough and mentally tough. And when I leave there, sometimes I will just sit in my car and have a good cry. (HSW)
**Actions performed at home**	
Walking	For me, I like to walk, so that’s…. We’ve got three little dogs who love to walk, so I’m making myself do a morning walk with them, which they all enjoy. And that way it just helps me clear my brain before I start work and I’m in a much better frame of mind. … On a weekend, I’ll do a two hour walk and just really expel it out. (RN/CE)
Leaving work at work	And then I try and bring no work home after 5:00. I’ll answer a phone but I won’t answer an email. (RN/CE)
Cooking	I cook, I create, doesn’t always work, but I just cut off from the rest of my dog and my husband and I just focus on cooking, so that is just my time. (CM)
Taking a long bath	For me, it’s about just when I have absolutely given everything to everybody, that’s when I need to go home and recharge and that might be a three-hour bath. (RN/CE)
Talking to spouse	I’ve got my partner that I’m able to talk to about it. He can empathise. He doesn’t understand it, but sometimes you just need to offload it in some way. (HSW)
**Actions performed by other people**	
When teams work well together	You do get some that you can’t connect with, but you can’t connect with everybody anyway, but the ones I work with now, we work really well together. We talk to each other. We don’t just say, “Well, I’m doing this and you’re doing that”. We all pitch in and we all help, yeah. (PCW)
When residents are happy	Having residents say that they loved me and that they enjoy being with me because I actually listen to them. I don’t just say, “I haven’t got time now. I’ll be back later”. … It makes me feel that I’m doing my job properly. (PCW)
When families give good feedback	I always feel good when I get feedback. Like, this morning we had a death over the weekend. A really long-term resident here. And the daughter drove in especially this morning to tell me what fantastic care she had. That makes me feel better, that what we’re doing is right. (RN/CE)
When someone thanks you	And you know, if somebody grabs your hand and say thank you, all the brickbats that you receive, that’s worth a thousand of them…. Just those kinds of things are really, that’s the thing that keeps you going. It’s not all the brickbats. It’s not even the bouquets. It’s those moments where you think I’ve made a difference in your life. That’s a good thing. You’ve made a difference in mine. (RN/CE)
When supervisors empathise	But when I needed support, my coordinator had the knowledge and understanding to listen and comfort me. She really listened. She just stopped. And she closed the door. And that was really important. … Because sometimes all you want to do is talk. You don’t need someone to solve anything for you. You just want to be heard. (LA)

## Data Availability

The datasets presented in this article are not readily available because approval for such disclosure was not part of the participant consent process.

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
