# Peer review of "“We Work in an Industry Where We’re Here to Care for Others, and Often Forget to Take Care of Ourselves”: Aged-Care Staff Views on Self-Care"

_geriatrics, 2025, doi:10.3390/geriatrics10010003_

Round 1

Reviewer 1 Report

Comments and Suggestions for Authors

I thank the authors for this opportunity to review their manuscript. This manuscript is on the “Aged Care Staff Views on Self-Care”. My comments are listed below:

Introduction:

First paragraph of introduction needs citation. This study was conducted during COVID-19 pandemic so there should be a paragraph in the introduction describing its potential influences. Aged care discussing about self-care was influenced by being in COVID-19 pandemic and now that we are post-pandemic the working context has changed so please provide the significance of discussing your results post pandemic.

Method:

Please specify in the body of the manuscript that : What are the “ELDAC marketing channels”? And more specifically: What kind of social media was used for recruitment? Since you did not collect demographics, the reader can understand how those recruitment channels could have influenced participation of aged care staff with various backgrounds.

Analysis:

Who were part of project advisory meeting? Pleases clarify.

Results:

Where does the participants’ quote starts? It is difficult to know as it is merged with the description of themes. The indention is not clear always. Using quotation mark can help distinguish quote from other parts.  

From the way you presented the results, it seems like the theme labeling comes from one participant interview. Theme labeling should reflect your synthesis of all the interviews put together under one theme. In other words, the label you chose for the themes should support the results you put underneath it. For example:  “Self-Care Is Important because There Are Many Challenging Aspects of Aged Care Work but It Often Gets Neglected” This theme is a complete sentence which needs to be cut short and really act as a label that represent everyone discussing this concept. It is not clear how is the quote that you have under this theme about impact of lockdown related to self-care?  Please provide a richer more in-depth analysis of your results for all of your themes and have shorter labelling.

Those who woke in home care may be different than those working in the residential system, therefore please specify whenever you can what was the location of the individuals discussing their self-care in the quotes.

Discussion:

The results should be discussed within the context of COVID and post-COVID. This study was conducted during pandemic to get insight on how aged care staff conduct self-care and many of the results may not be applicable after the pandemic which is a limitation of this study.

Comments on the Quality of English Language

Not applicable

Reviewer 2 Report

Comments and Suggestions for Authors

The comments for the authors are included in the attached document.

Comments on the Quality of English Language

Overall, the article makes a valuable contribution to the literature on self-care, especially in the challenging context of aged care.

The quality of the English language is generally good, with only minor edits needed to improve fluency and clarity in certain areas. I suggest a light review to correct minor inconsistencies and improve text cohesion, but there are no significant issues affecting comprehension.

Round 2

Reviewer 1 Report

Comments and Suggestions for Authors

Dear Authors,

Thank you for revising the manuscript. You have addressed many comments, some still needs to be addressed.

In your results, please provide a richer description of the analysis. The description is not strong enough. Also, two of your themes also do not reflect self-care:

1-Personalised Approaches to Self-Care and the Role of Workplace Relationship. It is not clear how is getting to know the colleague or debriefing self-care? Self-care are strategies implemented to improve health and well-being. From the way you presented the results, it is hard to see how this is related to self-care. 

You have this line "At work, the quality of relationships and interactions with colleagues and residents were frequently described as having an impact on how they feel". Self-care is very different than how they feel. 

2-Emotional Impact of Death and Dying: Coping Through Care, Connection, and Reflection. It is not clear how this is relevant to the focus of this work or self-care. Please explain in detail.

In your research, you had a range of aged care staff. How this influenced your results and things they said about self-care? A manager can think of self-care differently than an RN. This needs to be weaved into the results and the discussion. 

Comments on the Quality of English Language

N/A
